# A Personalized Approach to Improve Walking Detection in Real-Life Settings: Application to Children with Cerebral Palsy

**DOI:** 10.3390/s19235316

**Published:** 2019-12-03

**Authors:** Lena Carcreff, Anisoara Paraschiv-Ionescu, Corinna N. Gerber, Christopher J. Newman, Stéphane Armand, Kamiar Aminian

**Affiliations:** 1Laboratory of Kinesiology Willy Taillard, Geneva University Hospitals and University of Geneva, 1205 Geneva, Switzerland; stephane.armand@hcuge.ch; 2Laboratory of Movement Analysis and Measurement, Ecole Polytechnique Fédérale de Lausanne, 1015 Lausanne, Switzerland; anisoara.ionescu@epfl.ch (A.P.-I.); kamiar.aminian@epfl.ch (K.A.); 3Pediatric Neurology and Neurorehabilitation Unit, Department of Pediatrics, Lausanne University Hospital, 1011 Lausanne, Switzerland; corinna.gerber@chuv.ch (C.N.G.); christopher.newman@chuv.ch (C.J.N.)

**Keywords:** inertial sensors, gait detection, walking bout, personalization, cerebral palsy

## Abstract

Although many methods have been developed to detect walking by using body-worn inertial sensors, their performances decline when gait patterns become abnormal, as seen in children with cerebral palsy (CP). The aim of this study was to evaluate if fine-tuning an existing walking bouts (WB) detection algorithm by various thresholds, customized at the individual or group level, could improve WB detection in children with CP and typical development (TD). Twenty children (10 CP, 10 TD) wore 4 inertial sensors on their lower limbs during laboratory and out-laboratory assessments. Features extracted from the gyroscope signals recorded in the laboratory were used to tune thresholds of an existing walking detection algorithm for each participant (individual-based personalization: Indiv) or for each group (population-based customization: Pop). Out-of-laboratory recordings were analyzed for WB detection with three versions of the algorithm (i.e., original fixed thresholds and adapted thresholds based on the Indiv and Pop methods), and the results were compared against video reference data. The clinical impact was assessed by quantifying the effect of WB detection error on the estimated walking speed distribution. The two customized Indiv and Pop methods both improved WB detection (higher, sensitivity, accuracy and precision), with the individual-based personalization showing the best results. Comparison of walking speed distribution obtained with the best of the two methods showed a significant difference for 8 out of 20 participants. The personalized Indiv method excluded non-walking activities that were initially wrongly interpreted as extremely slow walking with the initial method using fixed thresholds. Customized methods, particularly individual-based personalization, appear more efficient to detect WB in daily-life settings.

## 1. Introduction

Wearable inertial devices have gained increasing popularity in recent years to monitor physical activity and gait-related parameters in real-life settings for clinical purposes [1,2]. Continuous and long-term assessment of gait parameters in the community (corresponding to gait ‘Performance’ as defined by the World Health Organization [3]) contributes to provide clinicians a better representation of a patient’s abilities in the context of every-day life [4]. The time spent walking, number of steps per day, maximum number of consecutive steps, mean walking speed, their variability and other ambulatory related-parameters during the day constitute precious data about a patient’s performance, and can inform about activity and participation [4]. However, accurate gait assessment in real-life settings is challenging [5]. Daily-life activities consist of a great variety of tasks, movements and body postures that are modulated in intensity and quality throughout the day, depending on a multitude of environmental and personal factors [6]. The detection of walking within this array of daily activities and also, the distinction between walking and non-walking types of locomotion (i.e., stair climbing and running) can prove complex.

One of the conventional methods for gait detection, i.e., walking bout (WB) detection, is based on the identification of consecutive steps, each step being identified from specific temporal gait events [7]. Gait is a periodic activity consisting of successive gait cycles, formed by two (left and right) steps. By convention in clinical gait analysis, the cycle is divided in two phases: The stance phase where the foot is in contact with the ground and the swing phase where it is not. Body segments’ acceleration and velocity have distinct properties during these two phases. Numerous gait recognition methods have used these specific features on accelerometer and/or gyroscope signals [8,9,10,11]. Commonly, peaks of shank angular velocity during the swing phase [8,12], or peaks of trunk or waist acceleration at foot contacts [5,13,14] are used to detect steps due to their simplicity [15], accuracy and repeatability [16]. In most methods, fixed thresholds are defined for peak detection, limiting the use of the algorithm for a specific population (mostly healthy adults or elderly people with or without disability [9,12,17,18]), and in specific settings (mostly laboratory settings). Therefore, the performance of such algorithms decreases when the gait patterns are unusual such as in slow and/or impaired walking [10,14,19,20] like in children with cerebral palsy (CP), and also when they are assessed in different settings [9,21] and footwear [22]. Adaptive thresholds throughout the assessment can also be set, giving the opportunity to adapt to the inter-steps variability within the monitoring [7,14,23]. One disadvantage of this type of approach is that every non-gait activity presenting similar periodicity (such as horse riding or legs swinging while seated) will trigger false step detection [19,22].

CP is a group of motor disorders resulting from damage of the developing brain, affecting about 1.8:1000 newborns [24]. Children with CP are characterized by heterogeneous impairments of the musculoskeletal system, with unilateral or bilateral body involvement. Only two thirds of children with CP can achieve community ambulation with or without mechanical walking aids (such as crutches, tripods or a walker), and independent walking is at the forefront of therapeutic objectives. In a previous study, we demonstrated that inertial sensors located on the shanks and thighs were able to accurately estimate gait parameters of children with CP and typical development (TD) in laboratory settings [25]. However, WB detection was not assessed in this study since the laboratory setting enabled the precise identification of the beginning and the end of the straight walking trials. There is thus a need to evaluate the performance of the system for WB detection in daily-life environments for this population. 

Given the heterogeneity of movement impairment in CP, the algorithms for WB detection and gait analysis based on fixed-thresholds approaches might not be optimal. The expected signal features for normal gait are attenuated or distorted in some cases, resulting in poor algorithm performance [21]. Therefore, an adaptation of signal processing thresholds to the individual’s movement features, or the so-called personalization, appears as a promising approach to improve algorithm’s performance. Personalization can be achieved through different ways, e.g., by adding input parameters such as anthropometric data or level of activity [26], by recognizing personal signal patterns [27], or by having a training phase to create personalized gait model using machine learning (ML) techniques [14,28]. To the authors’ knowledge, only one study tested an adaptive approach in children with CP and found acceptable accuracy using Random forest and support vector machine for walking detection [29]. The major drawback of pattern recognition and ML algorithms is the generalization of the model in different environments [29] and populations [27]. Furthermore, ML approaches need a high amount of data to train the model and are, therefore, difficult to achieve in the context of individual personalization [30]. Finally, no study was found emphasizing the effect of the quality of the walking detection on the distribution of a clinically meaningful parameter.

The aim of this study was therefore to evaluate if fine-tuning an existing WB detection algorithm [12] by various thresholds customized at the individual or group level could improve WB detection in children with CP and TD. The performance of the customized algorithms was assessed in terms of successful detection of WB. The clinical impact was assessed by quantifying the effect of an improved WB detection on the estimated walking speed distribution.

We hypothesized that tuning the algorithm based on the characteristics of the individuals’ movement pattern recorded in laboratory settings would improve the performance of WB detection in daily-life settings and have a significant impact on the walking speed estimation. 

## 2. Materials and Methods

### 2.1. Participants

Children and adolescents diagnosed with CP, and TD children were evaluated for this observational cross-sectional study. Participants were aged between 8 and 20 years. The inclusion criteria for the CP group were ability to walk in the community with or without mechanical aids, corresponding to levels of Gross Motor Function Classification System (GMFCS) I, II or III [31]. A balanced number of individuals per GMFCS levels was sought for this sample of participants in order to have a good representation of the heterogeneity in walking abilities. All participants provided written consent, and the protocol was approved by the hospital’s institutional ethical committee (CCER-15-176).

### 2.2. Protocol and Material

The participants were evaluated in a single measurement session which included two parts: (1) Laboratory assessments for thresholds set up, and (2) out-of-laboratory assessments for validation. During the whole measurement protocol, participants wore four synchronized inertial sensors (Physilog^®^4, GaitUp, Switzerland) fixed on each shank and each thigh. The sensors were safely fixed with hypoallergenic adhesive film (Opsite Flexigrid, Smith & Nephew Medical, UK). All sensors measured tri-axial acceleration and angular velocity at 100 Hz (range ±8 g, ± 2000 °/s).

Laboratory: Several straight gait trials were performed by the participants on a 20 m walkway, at 3 self-defined walking speeds: spontaneous, slow and fast. A total of 10 to 15 trials were recorded for each participant;Out-of-laboratory: A sequence of ‘daily-life-like’ activities was performed in the hospital corridors and hospital surroundings including static postures of lying, sitting and standing, walking on various surfaces (stones, grass, tarmac), along straight or curved trajectories, up and down stairs, and other free activities of the participant’s choice such as running, jumping or playing on swings. The sequence of activities lasted about 20 min and always started with a predefined body posture (lying on the back on a medical table). During the out-of-laboratory assessment, and for validation purpose, the evaluator was equipped with a camera (GoPro Hero+, USA) on his chest wearing a harness. The camera captured the participant during the entire sequence of daily-life-like activities.

### 2.3. Data Processing

The following sections will describe the data processing flow as shown in Figure 1.

#### 2.3.1. Pre-Processing

The inertial sensors and camera were manually synchronized by detecting the first lying posture on both the IMU signals and the video. Videos were labeled by the investigator using an open source software (BORIS) [32]. The type and timing of activities were exported in a csv file and were used as “ground truth” for WB validation. Data was processed and analyzed using MATLAB R2017 software (Mathworks, USA). IMU data recorded in laboratory was cut into trials. Out-of-laboratory IMU data was recorded continuously from the first lying posture, easily recognizable on the acceleration signals (constant value of about 1g during at least 3sec), until the end of the recordings. For calibration, principal component analysis was performed on the 3-axis angular velocity signals of each shank to align the pitch (around the medio-lateral axis) angular velocity (ω_z_Shank_) with the principal axis of movement during walking [33]. A high-pass infinite-responses (IIR) filter was applied to the signals to remove noise and cancel possible drift effect [12]. 

#### 2.3.2. WB Detection

As illustrated in Figure 1, three methods of WB detection were compared. The first, named ‘Init’, was the initial algorithm developed by Salarian et al. [12] using fixed thresholds to detect each step. The other two methods used the same algorithm but with thresholds customized at the population level, i.e., CP or TD, named ‘Pop’, or personalized at the individual level, named ‘Indiv’, using individual data obtained in Laboratory. Both approaches were tested to demonstrate the significance of considering individuals’ heterogeneity. The three methods are described in the following sections. 

##### *Initial* *Algorithm* *(Init)*

The method of Salarian et al. [12] was chosen since it has largely been described in research for various non-pathological (adults [12], elderly [34,35,36]) and pathological populations (patients with osteoarthitis [37], and Parkinson’s disease (PD) [12,38]), including studies with a large sample size (n > 800) [35,36]. Furthermore, it has recently been used in children with CP as a step counting reference in a semi-standardized setting [5]. It thus has the potential to be used for WB detection in the population of children with and without CP.

In Salarian et al.’s algorithm, the starting point of WB detection is the gait cycles detection on left and right side, through the identification of local maximum peaks on the recorded ω_z_Shank_ appearing around midswing (MS) [12], as illustrated on Figure 2. The peaks with an amplitude higher than 50°/s (Th1) are candidates for marking MS. If multiple peaks within 0.5 s (Th2) are detected, only the one with the highest amplitude is selected. If no consecutive MS of the same side are found within 1.5 s (Th3), the WB is ended, assuming that a gait cycle is always shorter than 1.5 s. A vector of MS times is then created for each side: one for the left (MS_L), and one for the right (MS_R) cycles. 

Once the cycles are identified, the alternation of right and left steps is controlled in order to define a WB. Starting with the first stride, the algorithm iterated over all (i) strides on both vectors MS_L(i) and MS_R(i) and compared the times to ensure that MS alternated between right and left sides. For the first two steps, if no successive MS_L(i) and MS_R(i) are found within 3.5 s (Th4), the WB is ended. Then, for the following steps, this threshold Th4 is adapted to the previous steps time intervals (Th4_adaptive_ = 1.5 s + mean steps time interval so far detected in the WB). 

Four fixed thresholds are thus considered in Salarian et al.’s original method: *Th1 = 50°/s:* Minimal amplitude of *MS*;*Th2 = 0.5 s:* Minimal time between *MS* of the same side;*Th3 = 1.5 s:* Maximal time between *MS* of the same side;*Th4 = 3.5 s:* Maximal time between *MS_R* and *MS_L*.

##### *Customized* *Algorithms*

Two customized methods were devised with the idea to tailor the four thresholds of the Init algorithm, from the gait trials obtained in laboratory for each individual (Indiv method) or each group CP or TD (Pop method). 

The laboratory gait trials at slow, spontaneous and fast speeds were thus used to extract the following features: minimum amplitude of MS (Th1), minimum (Th2) and maximum (Th3) time between MS of the same side, and minimum time between MS_R and MS_L (Th4) for each individual. This peak detection was made using the ‘findpeak’ function of MATLAB on the filtered ω_z_Shank_ and visually double-checked to ensure the accuracy and completeness of the MS detection (Figure 3). Criteria used to define these new thresholds are summarized for the Indiv and Pop methods in Table 1. Pop used the signal features of the population (i.e., CP or TD) to set the new thresholds. Indiv used the signal features of each individual, and each side (left and right) independently, to account for asymmetric gait patterns in the patients with CP.

To be as inclusive as possible, and knowing that all the signal characteristics extracted corresponded to true features, the extreme values were defined as the new thresholds. So, as detailed in Table 1, the minimal values found during laboratory trials were used to define the minimal amplitude of MS (*Th1*) and the minimal time between MS (*Th2*); and the maximal values were used to define the maximal time between MS (*Th3*) and the maximal time between MS_R and MS_L (*Th4*). For Indiv, *Th1* was defined as a percentage of the signal amplitude (95th percentile of *ω_z_Shank_*) since we presumed that the signal amplitude could differ between laboratory and out-of-laboratory gait. The defined thresholds were then used as inputs for the WB detection algorithm. 

#### 2.3.3. Walking Speed Computation 

Walking speed was computed using the double pendulum model, introduced by Aminian et al., from the shanks’ and thighs’ angular velocities [8]. The WB detected with each method (i.e., Init, Pop and Indiv), and with a duration of minimum 4 steps, were analyzed with the same gait analysis algorithm [8,12] for walking speed estimation. 

### 2.4. Analysis

Sensitivity (true positive rate), specificity (true negative rate), accuracy (true negative and positive rate) and precision (positive predictive value) were computed for each method against the video reference to evaluate their performance for WB detection. The values of these metrics vary between 0 and 1 (0 corresponding to the lowest performance and 1 corresponding to the highest performance). A tolerance of 2 s was applied to account for possible errors due to the manual synchronization between the sensors and camera. 

The customized method (Pop or Indiv) showing the higher sensitivity, specificity, accuracy and precision for WB detection was further compared with the initial method (Init) regarding the effect on walking speed estimation. Normality of the walking speed distributions were assessed using a MATLAB open access tool (Normality test package) [39]. Results were observed on the common basic statistics describing the distribution (mean, standard deviation, median, minimum, maximum, 1^st^ and 3^rd^ quartiles, Skewness and Kurtosis coefficients). An appropriate inference test was used to compare the distributions of walking speed estimated for the WB detected with Init and the selected customized method. The inference test was Wilcoxon unpaired tests in case of non-normal distribution or F-tests and T-tests in case of normal distribution. The level of significance was set at 0.05. Furthermore, Cumulative Distribution Function (CDF) plots were used to illustrate the effect of the chosen customized method as compared to the initial one. 

## 3. Results

### 3.1. Participants

Ten children with CP (4-GMFCS I, 3-GMFCS II, 3-GMFCS III) and 10 children with TD were included in this study. Details about their age, sex and clinical profiles are provided in Table 2.

### 3.2. Laboratory Gait Features

For each individual, gait features related to MS were extracted from pitch angular velocity signal, ω_z_Shank_, for straight walking performed at various speeds in laboratory setting. Mean, SD and range of these features were estimated for each CP and TD group (Table 3). 

### 3.3. WB Detection

Estimated sensitivity, specificity, accuracy and precision for WB detection using each method are indicated in Table 4. The method Init had maximum sensitivity (1.00), indicating that all actual WBs were detected by the algorithm. This was also the case for the customized methods with sensitivity values of 1.00 and 0.98 for Pop and Indiv, respectively. Specificity, accuracy and precision values were lower than the sensitivity for the method Init which means that some of the detected WB did not correspond to real WB. These performance metrics increased with method Pop, and even more with the method Indiv, for both groups and for the whole study population (Table 4). 

The most important improvement was observed for specificity, indicating that Indiv, provided better results for detection of true negatives, as illustrated in Figure 4.

In the illustrative example shown in Figure 4, the activity recorded between 650 s and 750 s, classified as ‘other activity’ on the reference video (green color), corresponded to playing on swings. The three methods detected MS during this activity because the amplitude of peaks was higher than the defined thresholds (*Th1*). However, unlike Init and Pop, the method Indiv managed to exclude most of these false positive WB, thanks to the adapted temporal thresholds between MSs.

### 3.4. Walking Speed Estimation

Since Indiv showed better performances for WB detection, this method was compared with the method Init regarding the walking speed distribution. The basic statistics describing the distribution are reported comparatively for the two methods and each participant in Table 5. Four illustrative examples of the difference between the walking speed distributions are represented using CDF plots in Figure 5. We noted that with the Indiv method, for all the participants except for one child with CP, a higher number of WB was detected, with a lower number of total gait cycles throughout the whole daily-life-like activity sequence. We observed a low impact of the personalized method on mean, standard deviation, median, maximum and quartiles of walking speed (up to 0.07 m/s difference overall). However, minimal walking speed changed of more than 0.4 m/s for half of the participants. We observed that the personalized method tended to bring the Skewness and Kurtosis coefficients toward 0, meaning that the distributions of speed with Indiv were more symmetric and normally tailed than with Init. Since walking speed distributions were not normally distributed, unpaired Wilcoxon tests were used to assess the significance of differences between distributions. Significant differences were found for seven participants (corresponding to 35% of our study population), four children with CP and three with TD. As well, for four other participants, the difference was close to the significance (*p* = 0.06) (that being 55% of the study population).

## 4. Discussion

This study aimed to evaluate if customized thresholds associated with methods using wearable inertial sensors can improve gait detection for children with or without CP. The main findings were that while both methods (Pop and Indiv) improved WB detection, Indiv provided best performance particularly in the detection of true negatives. Additionally, the clinical impact of such an improvement was investigated with the walking speed distribution over a 20-min sequence of daily-life-like activities. The results showed that the proposed individual-based personalized method had an effect on the distribution of walking speed for 35% of the participants. These findings could therefore meet clinicians since trustworthy data about the patient’s performances in daily life is required for therapeutic decisions.

The fixed thresholds of the initial method used in this study were designed and validated in laboratory settings for elderly people suffering from PD [12]. Two main reasons motivated us to evaluate the potential efficiency of new, more adaptive thresholds. First, children with CP demonstrate gait patterns that are much more unusual and heterogeneous than PD patients [40] and this was shown to be a criterion for decreasing performances of WB detection algorithms based on peak identification [10,20]. The second was the assessment environment which did not represent a daily-life situation. Wearable inertial sensors are meant to be used in daily-life environments where gait variability is higher [41]. Thus, validation of the system for the population of children with CP in out-of-laboratory settings was required. 

The proposed personalized method increased the number of detected WB as compared to the initial method (Table 5) since it cuts long WB detected by the Init method into several shorter ones, excluding hesitations and turnings. Furthermore, non-walking activities were more likely to be excluded from the analysis with the individual-based personalized method since the signal pattern (amplitude and/or timing of MS) was different from those of real gait trials. While high sensitivity is decisive to detect all WBs and to quantify the daily ambulatory activity, high specificity is necessary for effective qualification of gait impairments, i.e., computing relevant gait parameters. For example, walking cadence can be the forefront parameter for an individualized intervention in a natural environment [42]. In this case, estimated cadence should effectively correspond to actual walking cadence, and not to running or stair climbing cadence, to avoid misinterpretations [19]. The customized methods allowed us to improve the specificity while keeping high sensitivity; therefore, we can conclude that the system is relevant for both, assessment of physical behavior (e.g., WB duration, distribution and frequency) and gait impairment analysis (e.g., speed, cadence, stride length). Consequently, a positive impact on the reliability of single-day physical activity and walking performance, previously reported by Gerber et al. [43], could be expected. However, it is worth mentioning that to assess other aspect of gait such as endurance, continuous WBs including turnings and/or short breaks may be of interest. Since our proposed methods broke these WBs into many shorter WBs, gait endurance may be underestimated [44]. 

The method using thresholds customized at the population level (Pop) proved to be less specific and accurate than the personalization at the individual level (Indiv) (Table 4). This can be explained by the inter-subject variability in both groups, but especially in the group of children with CP. These results emphasized the relevance to consider not only the populations but the individuals to reach better performance of walking detection.

The distribution of computed gait parameters should be different if some false WB were excluded. This was mainly observed for the minimal walking speeds for most of participants, and actually half of them had more than 0.40 m/s difference. This can have a significant impact on the interpretations since the minimal clinically important difference regarding walking speed in CP is reported to be 0.10 m/s [45]. Furthermore, we previously found that the minimal detectable changes comparing walking speed estimation during two days of daily life in children with CP and TD were less than 0.22 m/s [43], meaning that a change of 0.40 m/s would have a clinical impact for a real life assessment. Unexpectedly, the customized methods were effective not only for children with CP but also for children with TD (Table 4). As far as the three patients with GMFCS III level are concerned, no significant difference was observed in walking speed distributions (Table 5), which may be explained by the fact that they chose to stay sitting on a bench during the ‘free activity of their choice’ because of the fatigue induced by the protocol’s activities. Indeed, the participants who benefitted the most from the customization of the method were those who performed unexpected activities during the ‘free activity of their choice’ part of the out-of-laboratory sequence such as playing on the swings. We expect that the differences might be more important for long-term real-life measurements, since the children would perform much more of these unanticipated activities (e.g., cycling, skating, horse-riding, skiing). Additionally, the impact on other gait parameters was not tested in this study but a greater difference could be expected especially for the knee range of motion since the knees flex and extend more during swinging on the swings than during walking.

The basic idea of customization proposed in this study was to fine-tune real-life gait analysis algorithm for an individual or a group of subjects by specific gait data obtained in laboratory from the same individual or group. This approach may be applied to many other situations where wearable systems are used to analyze abnormal gait or monitor activities such as postural transitions (sit-stand, stand-sit) in real-world situations, provided that the algorithm relies on specific signal features previously characterized in the laboratory controlled situation. Actually, in many situations, patients come to clinics for diagnosis, evaluations, regular checkups and/or therapy. These visits constitute good opportunities to record a few gait trials along a straight corridor path using inertial sensors and/or other instrument such as a camera or instrumented walkway, from which personalized thresholds or parameters of interest can be defined.

Many machine learning methods have been implemented for activity recognition or gait event detection and have shown good classification accuracy in (mostly healthy) adults [46,47]. Recently, a data-driven method for foot contact events was developed in children with various pathologies, including CP, and concluded that the accuracy of their approach was sufficient for most clinical and research applications in the pediatric population [48]. Ahmadi et al. also found acceptable accuracy for walking detection in children with a low level of CP using Random forest, support vector machine, and binary decision tree classifiers (between 90.3 and 96.5% in average) [29]. The accuracy achieved with the customized methods was within Ahmadi et al.’s range (90% for Pop and 95% for Indiv). The advantage of the method proposed in our study, with regard to the Ahmadi et al.’s method, is its personalization at the individual basis, thus its generalization to all type of pathologies. In addition, a notable strength of the individualization of the method through individual-based thresholds is the side independency. Indeed, in a population of unilaterally affected patients, distinguishing the more and less affected sides is relevant since the gait patterns can differ [22].

Although in this study the amplitude and timing of MS were the only values considered from the laboratory trials to tune the customized methods, the current findings underlined the potential for further, more sophisticated adaptive algorithms. Several additional characteristics, like the shape, amplitude or frequency of the signal, could be examined to further improve the customization. The signal pattern of the whole gait cycle could be used to detect gait cycles within long term monitoring by correlation [27]. Filtering often results in attenuation of the sharpness of specific signal features [49] so, in some cases, it can mask the information of interest. Personalized filters can improve gait event detection [5], especially in the CP population whose gait patterns can be noisy due to spastic movements, or drag feet [40]. Moreover, the proposed approach was designed for WB detection based on peak identification as proposed in the initial method [12]. Other methods could have been explored to adapt thresholds such as a moving window computing the root mean square (RMS) or the coefficient of variation of the signal to detect changes of signal amplitude [23]. Furthermore, a personalized approach has recently been proposed in a healthy population (30 young adults) using the signal itself to extract features (periodicity, energy, posture, etc.) to tune a step length model (online learning) [50]. This could be a relevant solution to test, to avoid the in-laboratory phase that we proposed for feature extraction. Indeed, good correlation was found between cadence and *Th2*, *Th3* and *Th4* (Appendix A). Instantaneous cadence roughly estimated by a method such as Fast Fourier Transform could thus directly determine the personalized thresholds, without any prior in-laboratory phase. Further work comparing the performance of such methods with the proposed personalized method should be undertaken.

The main limitation of this study was the modest sample size and the results should be confirmed with a larger population. Regarding the method, several limitations can be mentioned. The thresholds for both customized methods were defined to be as inclusive as possible since they corresponded to the extreme values of the overall supervised gait cycles at several walking speeds. First, various percentages of these extreme values could have been tested to determine if our choice was optimal. Second, we did not evaluate the repeatability of the definition of these thresholds even though intra and inter-day variability of gait exists, especially in patients with neurological disease. A future repeatability study could emphasize the performance of our approach. Next, the amplitude of the angular velocity peaks highly depends on the sensor alignment. The principle of axis alignment using PCA relies on the assumption that the principal axis of movement is the medio-lateral axis (sagittal plane). However, in children with CP with a high level of disability (GMFCS III), frontal and transverse components can be higher than normal, which may weaken the assumption for PCA alignment. In our case, the sensors were not removed between laboratory and out-of-laboratory assessments. However, since in real life the sensors may move among days, particular attention must be paid to the axis alignment. Finally, the interpretation of ‘spontaneous’, ‘slow’ and ‘fast’ walking may differ according to the participant and can lead to limited range of speed change, in laboratory trials, necessary to threshold fine-tuning and therefore decrease the performance of WB detection in real-life measurement settings.

## 5. Conclusions

Three methods of gait detection using inertial sensors were compared; the first with fixed thresholds [12], the second with population-based (CP-TD) thresholds, and the third with individual-based personalized thresholds. The third method was found to better improve WB detection, especially by excluding non-walking activities that were wrongly interpreted as extremely slow walking by the initial method. These findings are particularly relevant for real-life gait assessments to get a true representation of the patients’ performances. As far as the patient has the possibility to perform a quick in-laboratory or in-clinic gait measurement, the proposed approach is simple to set up, simple to implement and not specific to a pathology. 

## Figures and Tables

**Figure 1 sensors-19-05316-f001:**
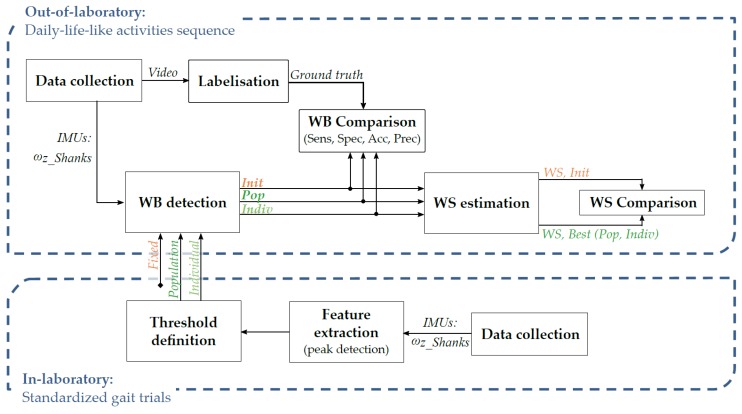
Workflow diagram. WB: Walking bouts; WS: Walking speed; Sens: sensitivity; Spec: specificity; Acc: accuracy; Prec: precision; ω_z_Shank_: pitch angular velocity of shanks.

**Figure 2 sensors-19-05316-f002:**
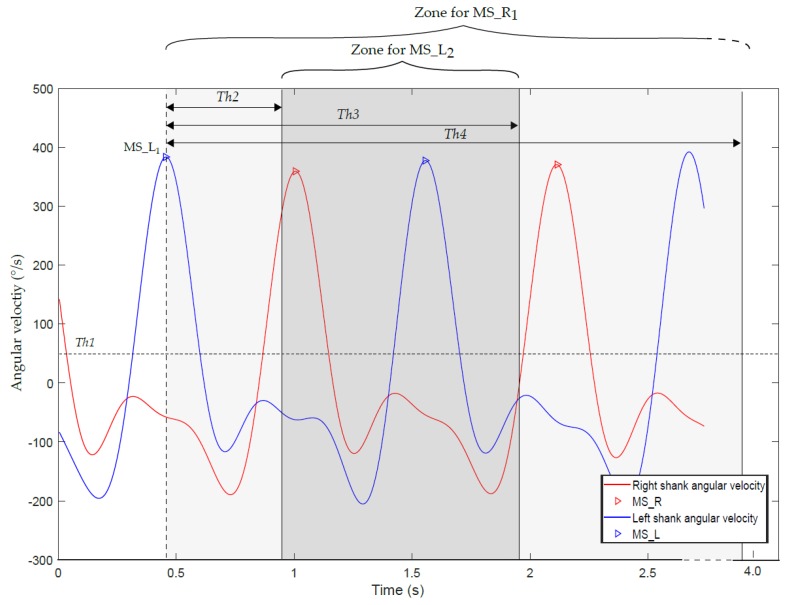
Illustration of the rules used in the initial algorithm. Two entire left cycles (in blue) and one entire right cycle (in red) are represented. MS_L: Left Midswing, MS_R: Right Midswing.

**Figure 3 sensors-19-05316-f003:**
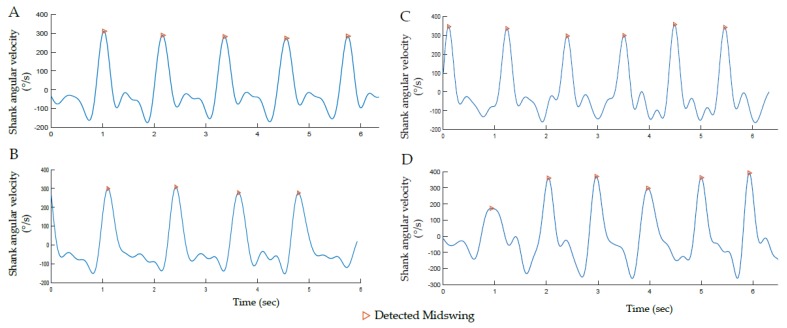
Illustrative examples of MS detection corresponding to maximum positive peaks on the pitch shank angular velocity signal. The figure illustrates signals recorded during laboratory gait assessment in four participants with different gait patterns: (**A**) TD child, (**B**) child with CP—GMFCS I, (**C**) child with CP—GMFCS II and (**D**) child with CP—GMFCS III.

**Figure 4 sensors-19-05316-f004:**
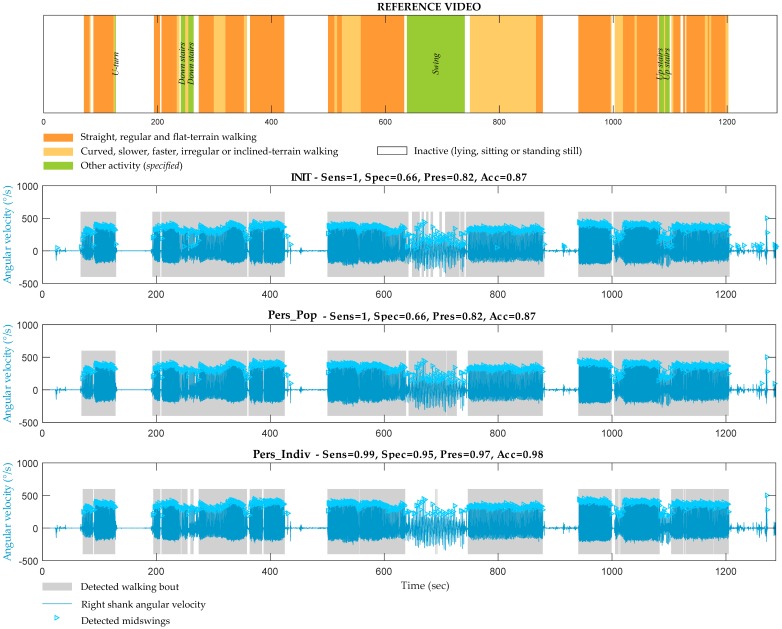
Example of walking bouts detection for one patient with CP by the three methods (Init, Pop and Indiv) in comparison with the reference. On the reference, orange strips correspond to straight, regular and flat-surface walking and yellow strips correspond to other type of walking such as slower, faster, on stones, on grass, on inclined-surfaces, along a curved trajectory. Green strips correspond to non-walking activities such as climbing or descending stairs, jumping, playing on the swings, running or playing around. The values of sensitivity (‘sens’), specificity (‘spec’), accuracy (‘acc’) and precision (‘prec’) are reported for each method. The gray strips corresponding to the detected walking bouts and the shank angular velocity signals with marks on detected midswing instants are presented superposed.

**Figure 5 sensors-19-05316-f005:**
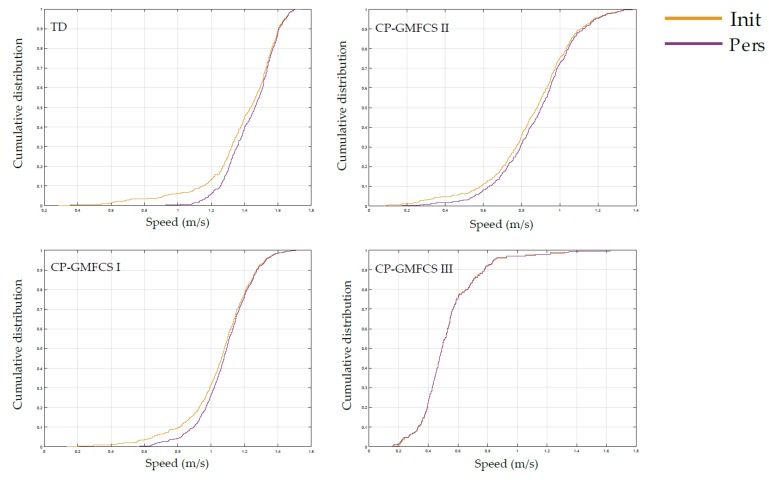
CDF plots of walking speed distributions resulting from Init and Indiv WB detection methods for a TD children and three CP patients with different GMFCS levels.

**Table 1 sensors-19-05316-t001:** Criteria for thresholds definition, based on the characteristics of the ω_z_Shank_ signal in laboratory for Pop and Indiv.

Threshold	Init	Pop		Indiv	
		CP	TD	Left Side	Right Side
*Th1* Minimal amplitude of MS (°/s)	50	Minimal MS amplitude in lab in the group	Minimal MS amplitude in lab in the group	($\frac{\mathrm{Minimal}\;\mathrm{MS}\_L\;\mathrm{amplitude}\;\mathrm{in}\;\mathrm{lab}\;}{{\mathrm{Max}\;\omega }_{z\_\mathrm{shank}}\mathrm{in}\;\mathrm{lab}\;}$) × 95th Percentile (ω_z_Shank_out lab)	($\frac{\mathrm{Minimal}\;\mathrm{MS}\_R\;\mathrm{amplitude}\;\mathrm{in}\;\mathrm{lab}\;}{{\mathrm{Max}\;\omega }_{z\_\mathrm{shank}}\mathrm{in}\;\mathrm{lab}\;}$) × 95th Percentile (ω_z_Shank_out lab)
*Th2* Minimal time between MS of the same side (s)	0.50	Minimal time between MS in lab in the group	Minimal time between MS in lab in the group	Minimal time between MS_L in lab	Minimal time between MS_R in lab
*Th3* Maximal time between MS of the same side (s)	1.5	Maximal time between MS in lab in the group	Maximal time between MS in lab in the group	Maximal time between MS_L max in lab	Maximal time between MS_R in lab
*Th4* Maximal time between MS_R and MS_L (s)	3.5	Maximal time between MS_R and MS_L in lab in the group	Maximal time between MS_R and MS_L in lab in the group	Maximal time between MS_R and MS_L in lab	

Init: Original algorithm; Pop: Population based method; Indiv: Individual based method; MS: Midswings; MS_L: Left midswings; MS_R: Right midswings; lab: Laboratory.

**Table 2 sensors-19-05316-t002:** Study participant details.

Group (CP/TD)	Sex	Age (years) Median [IQR]	Height (m) Median [IQR]	Weight (kg) Median [IQR]	GMFCS	Laterality	Orthosis	Walking Aids
TD (n = 10)	7 girls-3 boys	12.3 [11.5–13.6]	1.57 [1.52–1.62]	45.8 [40.0–56.8]	-	-	-	-
CP (n = 10)	6 girls-4 boys	13.0 [11.8–13.9]	1.56 [1.45–1.60]	43.5 [37.0–54.5]	4 GMFCS I-3 GMFCS II-3 GMFCS III	3 UCP-7 BCP	6 with AFO	1 with crutches-3 with walker *

UCP: Unilateral CP; BCP: Bilateral CP; AFO: Ankle Foot Orthosis; * including one patient using it for long distances only; IQR: Interquartile range.

**Table 3 sensors-19-05316-t003:** Population-based characteristics of the pitch angular velocity pattern extracted from laboratory gait assessments. The values in bold were used to define the thresholds for the Pop method.

	CP				TD			
Signal Features	mean	SD	min	max	mean	SD	min	max
Minimal amplitude of MS (°/s)	183	65	**109**	319	270	52	**193**	354
Minimal amplitude of MS (% of signal amplitude)	58	17	29	82	76	6	66	86
Minimal time between MS (s)	0.82	0.14	**0.64**	1.10	0.87	0.12	**0.67**	1.05
Maximal time between MS (s)	1.71	0.73	1.18	**3.53**	1.41	0.21	1.11	**1.92**
Maximal time between right and left MS (s)	0.93	0.38	0.60	**1.88**	0.72	0.11	0.56	**0.99**

MS: Midswings; SD: Standard deviation.

**Table 4 sensors-19-05316-t004:** Sensitivity, specificity, accuracy and precision of walking bouts detection for each method (based on the algorithm developed by Salarian et al. with fixed thresholds—Init, with population-based customized thresholds—Pop, and with individual-based personalized thresholds—Indiv), for each group and all the participants. Values are presented as medians [Interquartile range].

	Sensitivity			Specificity			Accuracy			Precision		
	Init	Pop	Indiv	Init	Pop	Indiv	Init	Pop	Indiv	Init	Pop	Indiv
TD	1.00	1.00	0.99	0.83	0.86	0.95	0.9	0.93	0.98	0.93	0.95	0.98
(n = 10)	[1.00–1.00]	[1.00–1.00]	[0.98–0.99]	[0.77–0.83]	[0.83–0.89]	[0.93–0.96]	[0.89–0.92]	[0.91–0.94]	[0.97–0.98]	[0.92–0.94]	[0.94–0.96]	[0.96–0.98]
CP	1.00	1.00	0.99	0.74	0.72	0.87	0.88	0.9	0.95	0.92	0.92	0.94
(n = 10)	[1.00–1.00]	[1.00–1.00]	[0.96–0.99]	[0.67–0.82]	[0.69–0.84]	[0.83–0.90]	[0.85–0.91]	[0.88–0.92]	[0.92–0.97]	[0.88–0.93]	[0.91–0.94]	[0.93–0.96]
**ALL**	**1.00**	**1.00**	**0.99**	**0.79**	**0.84**	**0.91**	**0.89**	**0.92**	**0.97**	**0.92**	**0.94**	**0.96**
**(n = 20)**	**[1.00–1.00]**	**[1.00–1.00]**	**[0.98–0.99]**	**[0.70–0.83]**	**[0.70–0.88]**	**[0.87–0.96]**	**[0.87–0.92]**	**[0.89–0.93]**	**[0.93–0.98]**	**[0.91–0.94]**	**[0.91–0.95]**	**[0.94–0.98]**

**Table 5 sensors-19-05316-t005:** Walking speed distribution descriptors for methods initial and personalized.

Group	GMFCS	WB (n)		Gait Cycles		Mean		SD		Median		Minimum		Maximum		1st Quartile		3rd Quartile		**Skewness**		**Kurtosis**		**Distribution Comparison ^w^ (*p*-Value)**
				(n)		(m/s)		(m/s)		(m/s)		(m/s)		(m/s)		(m/s)		(m/s)						
		***Init***	***Pers***	***Init***	***Pers***	***Init***	***Pers***	***Init***	***Pers***	***Init***	***Pers***	***Init***	***Pers***	***Init***	***Pers***	***Init***	***Pers***	***Init***	***Pers***	***Init***	***Pers***	***Init***	***Pers***	
TD	-	10	12	571	512	1.39	1.44	0.24	0.22	1.44	1.46	0.28	0.92	1.73	1.73	1.3	1.33	1.54	1.55	−1.8	−0.5	7.37	2.82	**0.02 ***
TD	-	6	13	611	553	1.19	1.22	0.23	0.21	1.23	1.24	0.23	0.64	1.67	1.67	1.09	1.12	1.32	1.33	−1.27	−0.41	6.40	3.54	0.06
TD	-	8	14	567	498	1.37	1.4	0.21	0.19	1.44	1.45	0.26	0.65	1.8	1.78	1.32	1.34	1.53	1.53	−1.81	−1.41	6.96	5.26	0.367
TD	-	7	10	589	543	1.14	1.16	0.17	0.16	1.16	1.17	0.18	0.64	1.47	1.43	1.07	1.08	1.24	1.24	−1.71	−0.34	9.64	3.48	0.31
TD	-	11	24	649	476	1.38	1.43	0.29	0.25	1.44	1.47	0.18	0.75	1.97	1.97	1.27	1.32	1.56	1.57	−1.5	−0.62	6.25	3.62	**0.041 ***
TD	-	7	14	602	543	1.29	1.31	0.2	0.19	1.31	1.32	0.14	0.45	1.71	1.68	1.21	1.22	1.41	1.41	−2.02	−1.31	11.40	7.17	0.321
TD	-	7	16	541	484	1.59	1.64	0.25	0.22	1.64	1.67	0.36	0.84	2.02	2.02	1.5	1.54	1.75	1.76	−1.68	−1.01	6.93	5.08	0.067
TD	-	7	14	506	465	1.3	1.34	0.14	0.13	1.35	1.36	0.29	0.68	1.57	1.57	1.27	1.29	1.41	1.42	−2.44	−1.61	10.56	8.01	0.061
TD	-	8	9	559	517	1.32	1.33	0.16	0.16	1.35	1.35	0.11	0.35	1.64	1.64	1.26	1.27	1.42	1.43	−2.38	−1.86	12.00	9.44	0.414
TD	-	10	23	610	540	1.28	1.32	0.26	0.22	1.32	1.35	0.31	0.54	1.98	1.98	1.16	1.21	1.43	1.44	−1.14	−0.65	5.24	4.71	**0.007 ***
CP	1	8	10	520	473	1.31	1.33	0.24	0.22	1.35	1.36	0.25	0.33	1.85	1.85	1.22	1.24	1.46	1.46	−1.31	−1.07	5.96	5.40	0.377
CP	1	15	25	694	606	1.05	1.09	0.22	0.2	1.08	1.09	0.14	0.55	1.51	1.51	0.97	0.99	1.18	1.19	−1.14	−0.4	5.31	3.49	**0.047 ***
CP	1	9	16	562	491	1.17	1.22	0.17	0.15	1.22	1.23	0.24	0.78	1.43	1.43	1.12	1.15	1.29	1.3	−2.05	−0.74	8.56	4.03	**0.016 ***
CP	1	7	18	607	532	0.9	0.93	0.25	0.22	0.9	0.92	0.16	0.58	1.57	1.5	0.78	0.81	1.02	1.04	0.09	0.612	3.99	3.28	**0.015 ***
CP	2	8	23	785	749	1.08	1.1	0.27	0.25	1.11	1.12	0.33	0.33	1.56	1.56	0.97	0.99	1.23	1.24	−0.75	−0.69	3.64	3.73	0.299
CP	2	6	23	646	601	0.94	0.98	0.18	0.16	0.97	0.98	0.16	0.30	1.5	1.5	0.87	0.9	1.05	1.06	−1.19	−0.44	5.57	5.37	**0.027 ***
CP	2	14	50	808	741	0.86	0.89	0.26	0.25	0.88	0.9	0.09	0.17	1.38	1.38	0.74	0.77	1.00	1.02	−0.74	−0.38	4.04	3.42	0.061
CP	3	21	20	378	347	0.53	0.53	0.18	0.19	0.49	0.49	0.17	0.16	1.63	1.63	0.41	0.41	0.59	0.59	1.82	1.73	8.69	8.44	0.991
CP	3	9	8	296	298	0.43	0.43	0.12	0.12	0.43	0.43	0.12	0.12	0.66	0.66	0.38	0.37	0.5	0.5	−0.35	−0.4	3.15	3.20	0.885
CP	3	17	18	817	783	0.66	0.68	0.1	0.11	0.68	0.68	0.14	0.37	1.00	1.00	0.63	0.63	0.74	0.74	−1.22	−0.24	5.78	3.91	0.707

Pers: Indiv (individual-based personalized method); ^w^: Unpaired Wilcoxon test for the comparison of the speed distribution between Init and Indiv. * indicate significant differences (*p* < 0.05) between methods. Gray gradient colors correspond gradually to the level of impairment of the participant. Negative and positive Skewness values mean that the distribution is skewed to the left and to the right respectively. Positive value of Kurtosis coefficient results from a tailed distribution whereas negative value corresponds to a flattened distribution.

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
