# Peer review of "A Personalized Approach to Improve Walking Detection in Real-Life Settings: Application to Children with Cerebral Palsy"

_sensors, 2019, doi:10.3390/s19235316_

Round 1

Reviewer 1 Report

This is an interesting paper with meaningful clinical implications. I have only two minor comments.

The abstract should be more explicit about the study aims. It should be clearly stated what aspects of the previous research are deficient. Walking speed or methods of measurement. the term “personalized” vs. “Population” based thresholding methods should be made more explicit. It is not very clear as to why setting up a threshold using individual data should be deviated from group average to such an extent that slow walking was wrongly detected in the previous study. A more detailed discussion is needed.

Reviewer 2 Report

The paper proposes an advanced method based on wearable inertial sensors to accurately detect walking bouts even in presence of abnormal gait.

The paper is well written and organized. The problem is widely studied in literature although the authors were able to highlight novel contribution, with promising results, provided the limitations discussed by the authors themselves.

The reviewer suggests to consider the following work in the state-of-the-art analysis: Using cloud-assisted body area networks to track people physical activity in mobility, BodyNets2015.

Another minor comment, which would be beneficial for the paper, is to include a workflow diagram that would help the reader to have an at-a-glance idea of the proposed processing method, from data collection up to detection.

Reviewer 3 Report

Authors evaluated the possibility to improve the performance of threshold-based algorithm for gait detection in CP patients by using personalization of threshold values. The paper is well-written and the aim of the paper is clearly stated.

However, two main issues appear and, in my opinion,  the manuscript cannot be considered for publication.

The main issue is related to the number of tested subjects. In fact, it is well-know that the gait parameters related are characterized by a not-negligible inter-subject variability. Thus, to validate and test an innovative approach only with six subjects is too restrictive for the publication of a journal article. It is more feasible for conference proceedings. The second issue is related to the absence of a repeatability analysis of the results. In fact, it is well-known the intra-day and inter-day variability of gait parameters, especially in patients with neurological diseases. For this reason, it is difficult to imagine the application of such approach in real life since a preliminary analysis in laboratory appears to be mandatory without the analysis of the repeatability of the personalized thresholds.

I strongly suggest authors to increase the number of subjects and to introduce an intra-day and inter-day repeatability analysis and resubmit a new version of the paper. After, I am glad to further review the paper.

Round 2

Reviewer 3 Report

Authors increased the number of subjects, improving the scientific quality of the manuscript.

Even though I strongly believe that the repeatability analysis is mandatory, I suggest to perform it as future work.